# R²AG: Rethinking Retrieval-Augmented Generation via Multimodal Coherence Understanding

## Abstract

While Multimodal Retrieval-Augmented Generation (MRAG) has shown promise in enhancing large language models, existing approaches are difficult to accurately capture the complex structural relationships between text and visual elements. This paper introduces R²AG (Rethinking Retrieval-Augmented Generation), a novel framework that extends MRAG to multimodal multi-level property graphs, significantly improving multimodal coherence understanding. Our approach represents multimodal content as interconnected nodes and edges in a property graph, capturing semantically rich relationships beyond conventional embedding distances. To address the exponential growth of graph complexity with additional hops, we propose an Implicit Chain-of-Thought (Implicit-CoT) technique that efficiently partitions and analyzes local subgraphs while deriving comprehensive features from both node semantics and structural properties. Additionally, we develop an improved graph matching algorithm that not only considers feature consistency but also recognizes semantic approximations and prioritizes rare entities, enhancing matching accuracy and robustness. Extensive experiments on public datasets demonstrate that R²AG outperforms state-of-the-art methods in multiple tasks requiring deep multimodal coherence understanding. Our code is available at `https://anonymous.4open.science/r/R2AG-4F58/`.

## 1 Introduction

In recent years, with the rise of GPT models Achiam et al. (2023), large language models (LLMs) have garnered significant attention from numerous researchers, emerging as a focal point of research in the field of Natural Language Processing (NLP). However, the performance of LLMs is often constrained by the knowledge acquired during the pre-training phase. To address this limitation, Retrieval-Augmented Generation (RAG) Lewis et al. (2020); Shuster et al. (2021) has emerged as a powerful new paradigm. RAG significantly enhances the capabilities of LLMs by dynamically retrieving and integrating external knowledge during the inference process, thereby improving factual accuracy and reducing hallucinations.

Currently, Retrieval-Augmented Generation (RAG) has achieved remarkable success in various text-based applications, including question answering and factual reasoning. Speculative RAG Wang et al. (2025) combines large general-purpose models with smaller specialized models, significantly enhancing both the accuracy and efficiency of RAG through parallel draft generation and efficient verification. GraphRAG Edge et al. (2024) extends this "draft-then-answer" philosophy with knowledge graph extraction, and approaching the problem from the perspective of integrating RAG with Query-Focused Summarization (QFS). Constructing entity knowledge graphs and community summaries substantially improves the comprehensiveness and diversity of GraphRAG. The subsequent HippoRAG Gutiérrez et al. (2024) further develops this approach by extracting graph information from knowledge bases, achieving even better results. However, these methods face challenges when processing rich multimodal content, leaving the full potential of RAG largely untapped. Extending RAG to multimodal content involves not only complex multimodal features but also intricate semantic relationships that transcend traditional text-based knowledge integration approaches, presenting unique challenges and opportunities.

The primary distinctions between traditional text-based RAG and Multi-modal RAG (MRAG) lie in both retrieval and generation processes. During the retrieval phase, text-modal RAG only needs to retrieve textual knowledge from document collections, whereas MRAG must integrate relevant knowledge and relationships across different modalities. In the generation phase, text-modal RAG only needs to output answers based on input text queries, while MRAG must utilize inputs and retrieved knowledge from various modalities to generate answers containing multi-modal information. Addressing these differences, M2RAG Ma et al. (2024) analyzes how humans naturally interact with multi-modal data in real-world scenarios, constructing multi-modal data preprocessing and joint embedding extraction strategies. Subsequently, it enhances generation based on multi-modal joint embeddings, achieving notable results. Contemporaneously, Seeing Beyond Chen et al. (2025) adopts a similar approach, achieving alignment between queries and knowledge through contrastive learning. To further enhance performance, it introduces an additional MLLM re-selection step, selecting the best matching knowledge from the top-k retrieval results of the alignment model. The subsequent EchoSight Yan & Xie (2024) also optimizes MRAG starting from text-image combination query relevance. However, these MRAG methods fail to consider the pervasive multi-level property graph structural relationships between text and images, instead utilizing only singular unified representations for feature space embedding distance calculations. In practical scenarios, this makes it difficult to effectively associate multi-level, multi-subject works with similar textual or image features, and also hinders Multimodal Coherence Understanding. This means that despite the excellent performance of multi-modal large language models (MLLMs) in understanding and generating textual and visual data, their ability to leverage structural information from multi-modal property graph relationships remains questionable.

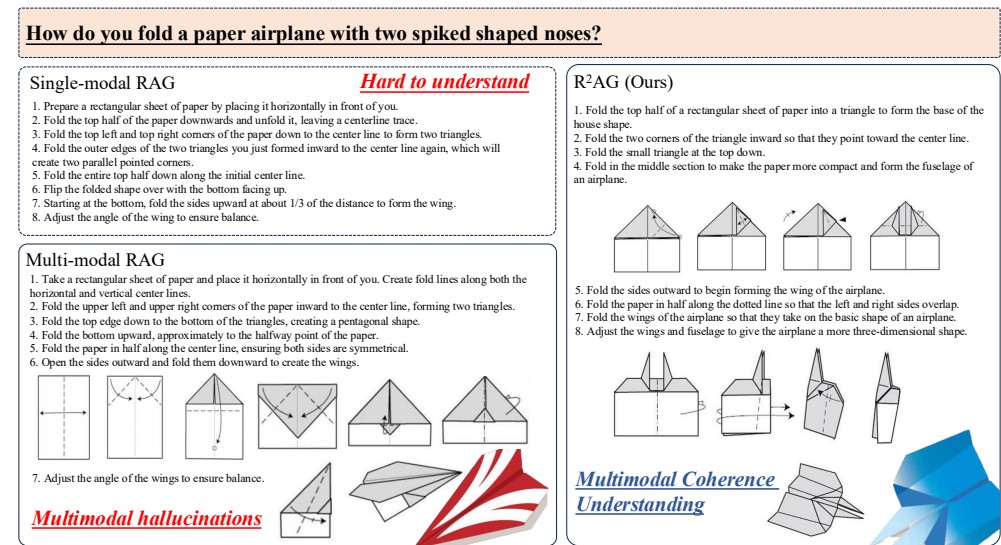

Figure 1: Test case using the latest advanced Single-modal RAG Method (left upper), Multi-modal RAG Method (left down) and R$^2$AG (right).

In this context, we extend multimodal retrieval-augmented generation to multimodal multi-level property graphs to enhance multimodal coherence understanding in multimodal large language models (MLLMs). For example, we conceptualize traditional advertising posters as having nodes in a multimodal information graph consisting of graphic elements and headline text, with shared visual styles serving as edges in this multimodal information graph. This structure uniquely connects each work with thousands of others in the graph, providing context beyond simple language descriptions or image references. However, as additional hops increase, the graph structure grows exponentially, resulting in excessively long context sequences. To address this, we employ an implicit chain-of-thought approach to partition and implicitly analyze local subgraphs, deriving comprehensive features from node text/image semantics and local graph structures, thereby reducing graph complexity and model hallucinations. Furthermore, traditional graph matching algorithms only consider feature consistency, struggling to capture synonyms, plurals, and spelling variations. Therefore, we

propose an improved graph matching algorithm for multimodal multi-level property graphs that considers both semantic approximation and assigns higher scores to rare entities. Experiments on public datasets demonstrate the effectiveness of our approach. As shown in Figure 1, this example expects to generate a paper airplane with two spiked-shaped noses. The Single-modal RAG approach cannot combine images and text, making it difficult to comprehend. While the Multi-modal RAG approach can process both images and text, it encounters multimodal hallucinations, incorrectly generating two spiked shapes in the final texture. Unlike those methods, $R^2AG$ can effectively provide an interactive answer with both images and text while avoiding multimodal hallucinations. Overall, our method makes several key contributions:

(1) **Construction of multimodal multi-level property graphs**: We extend multimodal retrieval-augmented generation to multimodal multi-level property graphs, aiming to achieve better multimodal coherence understanding through graph connectivity and node attributes. This structure provides context beyond simple descriptions by organizing multimodal information from different works (such as graphic elements and text) as nodes and edges.

(2) **Implicit Chain-of-Thought (Implicit-CoT)**: We propose an implicit chain-of-thought approach to partition and implicitly analyze local subgraphs, deriving comprehensive features from node text/image semantics and local graph structures. This approach not only reduces graph structural complexity but also decreases model hallucinations, enhancing detailed understanding of target node generation.

(3) **Improved graph matching algorithm**: We propose an enhanced graph matching algorithm for multimodal multi-level property graphs that considers not only feature consistency but also captures semantic approximations such as synonyms, plurals, and spelling variations, while assigning higher matching scores to rare entities. This innovation improves matching accuracy and robustness.

## 2 RELATED WORK

### 2.1 SINGLE-MODAL RAG

Retrieval Augmented Generation (RAG) Lewis et al. (2020); Shuster et al. (2021) can incorporate external knowledge to enhance the generation quality of Large Language Models (LLMs). Early researchers focused on addressing accuracy and efficiency issues in RAG, proposing various methods to optimize retrieval models Lee et al. (2021); Asai et al. (2024). Recently, numerous scholars have discovered that integrating knowledge graphs into RAG can significantly reduce hallucinations, subsequently proposing several knowledge graph-based enhancement approaches. These efforts have focused on several key areas: Graphs for Knowledge Indexing Sarthi et al. (2024); Wang et al. (2024b); Zhu et al. (2024), Knowledge Graph Construction from Corpus Cheng et al. (2024); Xu et al. (2024); Edge et al. (2024); Gutiérrez et al. (2024), and GraphRAG with Existing KGs Ao et al. (2025); Li et al. (2025); Zhu et al. (2025a); Sun et al. (2024a).

### 2.2 MULTIMODAL RAG

Multimodal Large Language Models (MLLMs) have become a research hotspot in the field of large language models in recent years due to their powerful multimodal processing capabilities Wang et al. (2024a); Tang et al. (2025); Shen et al. (2025). Early multimodal large language models conducted in-depth research on multimodal understanding. Flamingo Alayrac et al. (2022) utilized gated cross-attention layers to encode inputs and generate free-form textual outputs, achieving excellent image-text understanding. BLIP-2 Li et al. (2023) introduced the Q-Former architecture, mapping images into a hidden space aligned with text tokens in large language models Tang et al. (2024a). Taking this further, LLaVA Liu et al. (2023) simplified this framework through a projector and explored instruction tuning in the multimodal domain. In recent years, some researchers have focused on MLLMs' multimodal output capabilities. DreamLLM Dong et al. (2023) integrated LLM backbones with diffusion models Tang et al. (2024b) to achieve effective multimodal output generation. Emu2 Sun et al. (2024b) further expanded the parameter scale of this architecture, enhancing the in-context learning abilities of MLLMs. However, most existing methods overlook the relational dynamics between text and images, limiting their applicability to multimodal content generation tasks on multilevel multimodal attributed graphs.

To further enhance the multimodal content processing capabilities of large multimodal models, some researchers have proposed Multimodal RAG methods that can process both textual inputs and multimodal data. Initial Multimodal RAG methods similarly focused on multimodal content understanding, which limited their ability to provide users with rich multimodal responses Wu et al. (2025); Zhang et al. (2024). Subsequently, Zhu et al. Zhu et al. (2025b) proposed a Multimodal RAG method aimed at unified enhancement of both multimodal content understanding and generation capabilities. Later, M2RAG Ma et al. (2024) inherited this approach, further optimizing Multimodal RAG methods. However, overall, existing Multimodal RAG methods are still in their early stages and have not considered the multilevel attributed graph characteristics of multimodal data. They suffer from significant hallucination issues and struggle to achieve multimodal coherence understanding.

## 3 PRELIMINARY

### 3.1 MULTIMODAL MULTI-LEVEL PROPERTY GRAPHS (MMPGs)

A multimodal multi-level property graph can be defined as $\mathcal{G} = (\mathcal{V}, \mathcal{E}, \mathcal{I}, \mathcal{T})$, where $\mathcal{V}, \mathcal{E}, \mathcal{I}$, and $\mathcal{T}$ denote the sets of nodes, edges, images, and texts, respectively. Each node $v_i \in \mathcal{V}$ contains corresponding image information $i_{v_i} \in \mathcal{I}$ and textual information $t_{v_i} \in \mathcal{T}$. Taking the arts field as an example, MMPGs can construct rich representations of artist portfolios and movie poster networks. In an artist portfolio graph $(\mathcal{G})$, artist nodes $(v \in \mathcal{V})$ are interconnected through edges $(e \in \mathcal{E})$, with each node containing representative artwork images $(i \in \mathcal{I})$ and biographical text $(t \in \mathcal{T})$, showcasing influence relationships and stylistic heritage between artists. Similarly, in movie poster networks, film nodes connect not only to related entities such as directors and actors, but also integrate visual poster designs $(i \in \mathcal{I})$ with textual plot synopses $(t \in \mathcal{T})$, forming a multidimensional information structure. These multimodal attributed graphs create semantically rich knowledge representation frameworks by merging visual and textual features, providing a more comprehensive and in-depth foundation for applications such as artwork analysis and film recommendations.

### 3.2 MULTI-MODAL RETRIEVAL BASED ON MMPGs

Here we introduce multi-modal retrieval based on MMPGs, which retrieves the most relevant multimodal content for each user query $Q$ by processing a multimodal dataset $\mathcal{K} = \{\mathcal{G}_1, \cdots, \mathcal{G}_n\}$ containing structured relationships. Each graph $\mathcal{G}_i = (\mathcal{V}_i, \mathcal{E}_i, \mathcal{I}_i, \mathcal{T}_i)$ consists of a set of nodes $\mathcal{V}_i$, edges $\mathcal{E}_i$, images $\mathcal{I}_i$, and texts $\mathcal{T}_i$, which are interconnected through semantic and structural relationships. The retrieval process first extracts multimodal features from the query $f_Q = \phi\left(f_v\left(Q^{vis}\right), f_t\left(Q^{text}\right)\right)$, and then performs graph matching to identify the most relevant structures. The graph matching mechanism considers both content similarity and structural correspondence by calculating the similarity between the query graph $\mathcal{G}_Q$ and the target graph $\mathcal{G}_t$ : $\text{sim}\left(\mathcal{G}_Q, \mathcal{G}_t\right) = \alpha \cdot \text{ContentSim}\left(f_Q, f_t\right) + (1 - \alpha) \cdot \text{StructSim}\left(A_Q, A_t\right)$, where $A_Q$ and $A_t$ represent the adjacency matrices of the query and target respectively, and $\alpha$ is a balancing parameter. Finally, we will get retrieved Documents $\mathcal{D} = \{\mathcal{D}_1, \cdots, \mathcal{D}_n\}$.

### 3.3 MULTI-MODAL GENERATION BASED ON MMPGs

In a multimodal multi-level property graph $\mathcal{G} = (\mathcal{V}, \mathcal{E}, \mathcal{I}, \mathcal{T})$, where $\mathcal{V}$ represents the set of nodes, $\mathcal{E}$ represents the set of edges, $\mathcal{I}$ and $\mathcal{T}$ represent the sets of images and texts respectively, given a node $v_i \in \mathcal{V}$ within graph $\mathcal{G}$, our goal is to generate a semantically consistent and complementary image-text pair $(i_{v_i}, t_{v_i})$ by learning a mapping function $f : \mathcal{V} \times \mathcal{G} \rightarrow \mathcal{I} \times \mathcal{T}$, such that the generated content maintains both structural consistency and cross-modal semantic alignment, which has extensive applications in knowledge graph visualization, social media content recommendation, and other domains.

## 4 METHODOLOGY

After establishing the fundamental theoretical groundwork, we will now provide a concise overview of the entire $R^2AG$ framework, introducing our key components and specific implementation methods for multimodal coherence understanding. The overall framework is illustrated in the Figure

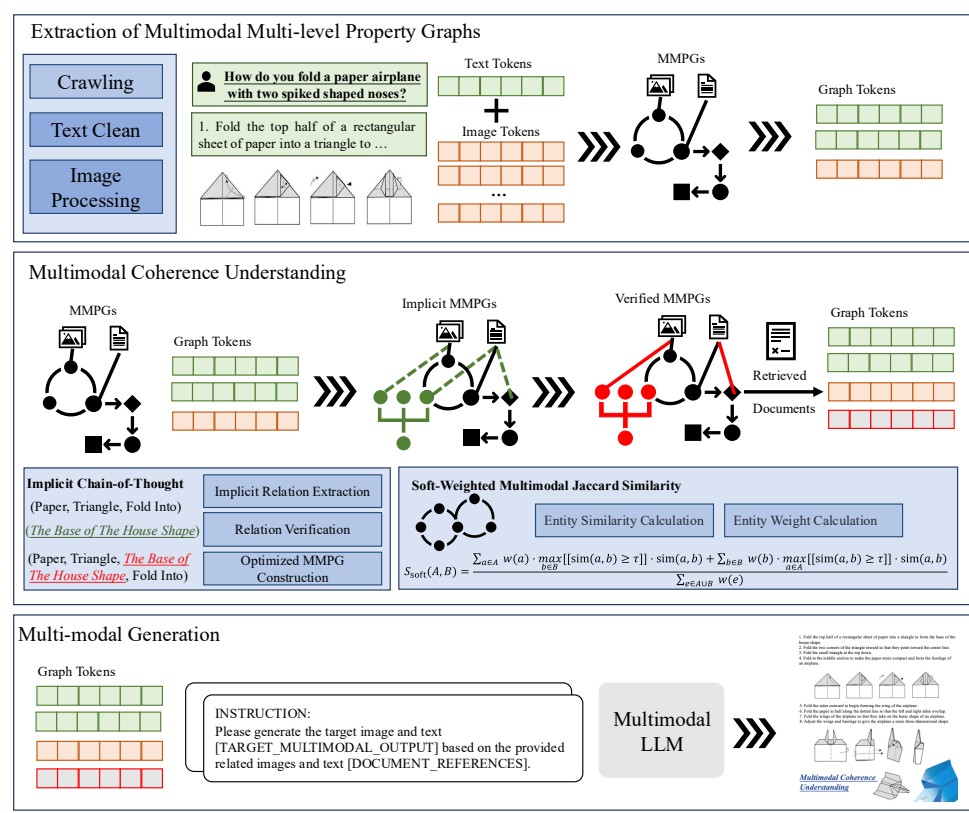

Figure 2: The overall framework of our proposed multimodal RAG method named R$^2$AG, which includes Extraction of Multimodal multi-level property graphs, Multimodal Coherence Understanding and Multimodal Generation.

2. We begin by implementing the Extraction of Multimodal Multi-level Property Graphs. Subsequently, we employ Implicit Chain-of-Thought reasoning to estimate the subgraph hierarchy and positioning relevant to the current query, and through Implicit Thinking, we generate derivative graphs based on the current query and estimated subgraphs. It is important to note that the derivative graphs generated during this process are only temporarily stored in the graph database for the current query, thus not affecting the complexity of subsequent retrieval operations. Following this, we have constructed an enhanced graph matching methodology to implement the final derivative graph-based matching, ultimately determining the definitive subgraph. Based on these final subgraphs, we have developed a multistage generation approach to facilitate comprehensive multimodal content generation.

## 4.1    EXTRACTION OF MULTIMODAL MULTI-LEVEL PROPERTY GRAPHS

To implement the construction of Multimodal Multi-level Property Graphs (MMPG), we first execute rigorous Text Cleaning and Image Processing on the crawled heterogeneous corpus. For Text Cleaning, we implement a comprehensive NLP pipeline incorporating noise and artifact elimination, UTF-8 character normalization, regex-driven whitespace optimization, and formatting standardization. This process involves TF-IDF vectorization for metadata filtration, Levenshtein distance-based typographical correction, and dependency parsing for ensuring cross-corpus syntactic consistency. The resulting preprocessed text maintains semantic fidelity while being optimized for downstream semantic role labeling and entity disambiguation tasks. For Image Processing, we deploy state-of-the-art computer vision techniques for quality enhancement, dimensional normalization, and artifact remediation. Our image preprocessing pipeline incorporates wavelet-based adaptive denoising, histogram equalization for contrast enhancement, bicubic interpolation for resolution standardization,

and deep image matting for background normalization. These rigorously processed images function as high-fidelity visual nodes within our property graph topology, facilitating robust multimodal feature extraction and relationship modeling.

Following these preprocessing operations, we implement a sophisticated retrieval-augmented generation framework based on the user's query $Q$. We conduct preliminary retrieval to obtain a set of pertinent documents $\mathcal{D} = \{\mathcal{D}_1, \cdots, \mathcal{D}_n\}$. Subsequently, we employ a multi-encoder architecture to transform the user's query $Q$ and textual components from the retrieved documents $\mathcal{D}$ into dense semantic vector representations via our text tokenization pipeline. Concurrently, we process the visual elements within $\mathcal{D}$ through our vision transformer encoder to generate corresponding image token embeddings. These multimodal representations undergo cross-attention mechanisms and feature alignment to establish a heterogeneous property graph structure, where nodes represent semantic units and edges capture cross-modal relationships. This graph topology facilitates complex reasoning through subgraph traversal algorithms and attention-based message passing, enabling effective integration of multimodal information for downstream inference tasks.

## 4.2 MULTIMODAL COHERENCE UNDERSTANDING

To achieve superior Multimodal Coherence Understanding, we employ an Implicit Chain-of-thought methodology to deeply optimize the generated MMPGs (Multimodal Multi-level Property Graphs). Building upon these optimized MMPGs, we propose a novel Soft-Weighted Multimodal Jaccard Similarity metric to measure graph similarity between different MMPGs, enabling more effective document retrieval. MMPGs serve as powerful representations for complex relationships across diverse data modalities, yet conventional approaches often fail to capture implicit connections that require sophisticated reasoning. In this paper, we introduce a novel Implicit Chain-of-Thought (ICoT) methodology that significantly enhances MMPGs by discovering, verifying, and incorporating latent relationships through advanced language model reasoning. Our approach unlocks previously inaccessible graph structures that substantially improve cross-modal retrieval performance.

**Step 1: Implicit Relation Extraction:** We employ an Implicit Chain of Thought (ICoT) approach to identify potential relations in MMPG. Unlike explicit relations that can be directly extracted from context, implicit relations require complex multi-step reasoning. We provide the original multimodal context $Q$ along with previously generated explicit relation set $\mathcal{E}_0$ to guide the LLM in discovering hidden connections within the graph structure. The ICoT reasoning process identifies intermediate relation paths $T_i = (v_i, v_{i+1}, e_i)$ that form coherent chains, where $v_i, v_{i+1} \in \mathcal{V}$ are nodes and $e_i$ represents the relation type between them. These chains can generate higher-order relation triplets $T_{1 \to n} = (v_1, v_n, e_k)$ derived from sequential reasoning paths $T_1, T_2, \ldots, T_{n-1}$, where $n$ denotes the chain length. This process can be formalized as:

$$\mathcal{E}'_i = \bigcup_{i,j \in \mathcal{V}} \{(v_i, v_j, e_k)\} \tag{1}$$

where $e_k$ represents $k$ different relation types inferred through the ICoT reasoning process.

**Step 2: Relation Verification:** Leveraging the self-verification capabilities of pre-trained LLMs, we implement a sophisticated verification mechanism to evaluate the validity of each relation within the MMPG context. The LLM acts as a discriminative agent, assigning confidence scores to each proposed relation triplet based on multimodal consistency. For each triplet, the evaluating agent generates a confidence value $\mathcal{V}(i, j, k)$ indicating the likelihood that relation $e_k$ correctly connects nodes $v_i$ and $v_j$. We establish an adaptive threshold $v_{th}$ to filter relations:

$$\mathcal{E}_i = \bigcup_{i,j \in \mathcal{V}, \mathcal{V}(i,j,k) \geq v_{th}} \{(v_i, v_j, e_k)\} \tag{2}$$

This LLM-driven verification process significantly enhances the coherence of the MMPG by eliminating spurious connections while preserving semantically meaningful relations.

**Step 3: Optimized MMPG Construction:** The final optimized MMPG incorporates both explicit and implicit relation sets into a comprehensive graph structure. We formulate this as a conditional probability maximization problem:

$$\mathcal{G} = \arg\max_{\mathcal{G}_i} P\left(\mathcal{G}_i \mid \mathcal{V}, \mathcal{E} = [\mathcal{E}_e, \mathcal{E}_i], \mathcal{I}, \mathcal{T}, Q\right) \tag{3}$$

where $\mathcal{E}_e$ represents the set of explicit relation edges, $\mathcal{E}_i$ represents the set of implicit relation edges discovered through ICoT reasoning, $\mathcal{V}$ is the set of nodes, $\mathcal{I}$ contains image information, and $\mathcal{T}$ contains textual information. This optimized MMPG construction serves as the foundation for our novel soft-weighted multimodal Jaccard similarity measure(which is detailed in A), enabling more effective cross-modal document retrieval by capturing semantic and structural similarities between complex multimodal graphs.

Table 1: Comparison of subjective indicators in textual dimension

| Method | Evaluator | Flu. ($\uparrow$) | Rel. ($\uparrow$) | CP. ($\uparrow$) | Faith. ($\uparrow$) |
|---|---|---|---|---|---|
| MuRAR | GPT-4 | 76.8/100 | 77.5/100 | 76.5/100 | 76.2/100 |
| | Human | 72.7/100 | 73.3/100 | 72.3/100 | 72.8/100 |
| EchoSight | GPT-4 | 78.5/100 | 79.8/100 | 78.4/100 | 77.9/100 |
| | Human | 74.6/100 | 75.2/100 | 74.1/100 | 74.5/100 |
| M2RAG | GPT-4 | 80.6/100 | 79.4/100 | 82.4/100 | 79.6/100 |
| | Human | 76.8/100 | 74.6/100 | 78.0/100 | 75.8/100 |
| **R$^2$AG (Ours)** | GPT-4 | **88.2/100** | **89.2/100** | **87.2/100** | **83.0/100** |
| | Human | **82.2/100** | **85.2/100** | **81.6/100** | **81.4/100** |

Table 2: Comparison of subjective indicators in visual dimension

| Method | Evaluator | Coher. ($\uparrow$) | Help. ($\uparrow$) | Ref. ($\uparrow$) | Recall ($\uparrow$) |
|---|---|---|---|---|---|
| MuRAR | GPT-4 | 74.8/100 | 75.6/100 | 74.5/100 | 74.2/100 |
| | Human | 70.9/100 | 71.4/100 | 70.3/100 | 70.8/100 |
| EchoSight | GPT-4 | 76.8/100 | 77.9/100 | 76.5/100 | 76.2/100 |
| | Human | 72.7/100 | 73.3/100 | 72.3/100 | 72.8/100 |
| M2RAG | GPT-4 | 80.2/100 | 81.2/100 | 80.2/100 | 79.6/100 |
| | Human | 76.4/100 | 77.0/100 | 76.0/100 | 76.2/100 |
| **R$^2$AG (Ours)** | GPT-4 | **87.0/100** | **88.0/100** | **84.6/100** | **85.0/100** |
| | Human | **83.2/100** | **83.8/100** | **82.0/100** | **80.0/100** |

### 4.3 Multimodal Generation

After obtaining the final retrieved document collection, we implemented a multi-stage multimodal output generation system based on large language models. This architecture consists of three key processing stages:

(1) Semantic Content Initialization: The system first generates a coherent foundational text representation based on multimodal input resources, expressed as $T_0 = G_{\text{init}}(\mathcal{D}, \mathcal{I})$, where $\mathcal{I}$ represents the retrieved image collection;

(2) Visual Information Embedding Decision: Using an intelligent segmentation algorithm to structurally divide the initial text $T_0 \rightarrow \{S_1, S_2, ..., S_n\}$, and determining the optimal set of image embedding positions $P = \{p_1, p_2, ..., p_k\}$ through MLLM;

(3) Multimodal Content Integration and Optimization: Purposefully fusing visual elements with textual content and refining each text paragraph through semantic enhancement algorithms, resulting in $T_{\text{refined}} = R(T_0, P, \mathcal{I})$.

Upon completion of processing, the system performs tokenization on the refined text, generating structured Graph tokens $G = \tau(T_{\text{refined}})$, and ultimately constructs specialized instructions:

$$f(\mathcal{D}, \mathcal{I}) \rightarrow \mathcal{O}_m \tag{4}$$

Where $\mathcal{D}$ and $\mathcal{I}$ represent the retrieved relevant texts and images respectively, and $\mathcal{O}_m$ represents the target multimodal output satisfying $\text{sim}(\mathcal{O}_m, \{\mathcal{D}, \mathcal{I}\}) \geq \theta$, with $\theta$ being the similarity threshold. This is then input into a high-performance multimodal large language model, achieving the perfect

combination of precise information expression and visual presentation. This process essentially involves learning a mapping function similar to $f : \mathcal{V} \times \mathcal{G} \to \mathcal{I} \times \mathcal{T}$ as described in the preliminary, ensuring that the generated content maintains both structural and semantic consistency.

Table 3: Average objective indicators across all five multimodal tasks

| Method | FVD ($\downarrow$) | CLIP Score ($\uparrow$) | DOVER ($\uparrow$) |
|---|---|---|---|
| MuRAR | 20.26/26.23 | 17.68/19.02 | 60.15/66.32 |
| EchoSight | 20.08/23.74 | 19.15/20.03 | 62.03/65.12 |
| M2RAG | 18.02/21.82 | 22.88/23.02 | 64.44/67.62 |
| **R$^2$AG (Ours)** | **12.45/15.87** | **26.74/27.04** | **68.14/71.22** |

## 5 EXPERIMENT

### 5.1 EXPERIMENTAL SETUP.

To validate the superiority of our proposed R$^2$AG method, we compare the generated output against the state-of-the-art (SOTA) methods. For this purpose, we have selected the previous generation SOTA model, M2RAG Ma et al. (2024), as our benchmark. To ensure a fair evaluation, we used the data mentioned in the M2RAG Ma et al. (2024) as our test data, and abstracted the specific tests into five classic tasks which primarily include: Visual Question Answering, Multimodal Understanding, Multimodal Generation, Cross-modal Retrievaland Multimodal Dialogue. Meanwhile, to ensure fine-grained performance analysis across different tasks, we eliminated data not belonging to these tasks and supplemented the Visual Question Answering category, which had fewer samples, ultimately obtaining 1,200 test data. These tasks require models to simultaneously process and associate different types of information, achieving cross-modal understanding and generation capabilities. Ultimately, we compared our method with M2RAG, benchmarked its performance based on QwenWang et al. (2024a), and conducted extensive comparisons using both subjective and objective metrics.

**Subjective indicators.** In evaluating multimodal generation systems, we employ a comprehensive set of metrics to measure model performance. **Fluency (Flu.)** assesses the linguistic quality of generated text, including grammatical correctness, syntactic structure reasonability, and expression coherence; **Relevance (Rel.)** measures the degree of match between content and query, reflecting the model's ability to understand user intent; **Context Precision (CP.)** evaluates the model's accuracy in integrating dialogue history information, assessing its effectiveness in handling long-term dependencies; **Faithfulness (Faith.)** examines the accurate expression of factual information, quantifying the model's ability to avoid "hallucinations." In the visual dimension, **Coherence (Coher.)** evaluates the internal consistency of visual output and the semantic mapping accuracy with textual descriptions; **Helpfulness (Help.)** measures the practical contribution of visual output to meeting user needs, reflecting information value; **Reference Quality (Ref.)** assesses the visual similarity and quality comparison between generated images and professional standards; **Recall** quantifies the completeness of model in capturing all key visual elements in user queries. These metrics constitute a multi-dimensional evaluation framework that can both independently measure specific capabilities and be weighted to form comprehensive scores, providing theoretical foundation and quantitative basis for iterative optimization of multimodal systems. We utilized prompt engineering to enable GPT-4 to achieve fair evaluation across different metrics, while ensuring standard consistency alignment among human raters.

**Objective indicators.** For the evaluation of Multimodal output, we treat multimodal outputs as video streams, applying established video assessment methodologies. Our framework employs three metrics: **FVD** quantifies authenticity by comparing synthetic and real output distributions via pretrained network features; **CLIP score** measures text-visual alignment through cosine similarity in vision-language space; and **DOVER** assesses both content fidelity and descriptive coherence using spatial-temporal fusion and domain-aware contrastive learning.

## 5.2 COMPARISON WITH ADVANCED METHODS.

The results presented in the Table 1, Table 2 and Table 3 demonstrate a clear superiority of the $R^2AG$ method over M2RAG across both subjective and objective metrics in various dimensions. In the textual dimension, $R^2AG$ outperformed M2RAG in all evaluated indicators, particularly in fluency and relevance, with GPT-4 scores of 88.5 and 89.3, respectively, compared to M2RAG's 80.5 and 79.5. Human evaluators echoed this sentiment, highlighting $R^2AG$'s enhanced text generation quality. Similarly, in the visual dimension, $R^2AG$ excelled with GPT-4 scores of 87.0 for coherence and 88.0 for helpfulness, significantly higher than M2RAG's scores of 80.2 and 81.2. Human assessments further corroborated $R^2AG$'s superior performance across all visual metrics, indicating improved visual information conveyance. Lastly, in the objective evaluation of multimodal tasks, $R^2AG$ achieved a Fréchet Video Distance (FVD) of 12.45, markedly lower than M2RAG's 18.02, alongside higher scores in CLIP and DOVER metrics. Overall, these findings underscore $R^2AG$'s remarkable advancements in generation quality and information transmission capabilities across both textual and visual modalities.

## 6 CONCLUSION

In this paper, we presented **$R^2AG$**, a novel approach rethinking retrieval-augmented generation through enhanced multimodal coherence understanding. Our framework addresses existing RAG limitations by capturing intricate structural relationships between modalities through three key contributions: multimodal multi-level property graphs for rich contextual information, an Implicit Chain-of-Thought approach reducing hallucinations, and an improved graph matching algorithm prioritizing rare entities. Comprehensive evaluations across five multimodal tasks demonstrate our significant performance improvements over state-of-the-art methods. By rethinking how retrieval-augmented generation can be enhanced through structured multimodal reasoning, $R^2AG$ bridges the gap between traditional RAG systems and the complex requirements of real-world multimodal applications.

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

## A  APPENDIX

### A.1  DETAILED PROMPT OUTPUT OF TEXTUAL DIMENSION EVALUATION

> Please evaluate the multimodal system output according to the following criteria, with a total score of 100 points: **Fluency (Flu.):** Assess the grammatical correctness, syntactic structure, and coherence of the text.*Score: Justification:*
>
> **Relevance (Rel.):** Measure how well the content addresses the user's intent and includes key information while avoiding irrelevant content. *Score: Justification:*
>
> **Context Precision (CP.):** Evaluate the accuracy in referencing previous conversation information and maintaining topical coherence. *Score: Justification:*
>
> **Faithfulness (Faith.):** Examine the factual basis of the content, avoiding fabricated information and accurately expressing uncertainty about knowledge boundaries. *Score: Justification:*
>
> **Overall Assessment:**

This section presents our textual evaluation framework for multimodal outputs. Using four dimensions (Fluency, Relevance, Context Precision, and Faithfulness), evaluators score each aspect and provide justifications. Together, these metrics create a 100-point assessment system that offers quantifiable data for system improvement. Above is the evaluation format used throughout our analysis.

### A.2  DETAILED PROMPT OF VISUAL DIMENSION EVALUATION

This section presents our visual evaluation framework for multimodal outputs using four dimensions: Coherence, Helpfulness, Reference Quality, and Recall. Evaluators score each aspect with justifications. Coherence assesses visual consistency with text, Helpfulness measures practical utility, Reference Quality compares against professional standards, and Recall evaluates completeness. Together, these form a 100-point system providing insights for improvement. Our evaluation format follows:

> Please evaluate the multimodal system output according to the following visual dimension criteria, with a total score of 100 points:
>
> **Coherence (Coher.):** Evaluate the internal consistency of visual output and the semantic mapping accuracy with textual descriptions. *Score: Justification:*
>
> **Helpfulness (Help.):** Measure the practical contribution of visual output to meeting user needs, reflecting information value. *Score: Justification:*

**Reference Quality (Ref.):** Assess the visual similarity and quality comparison between generated images and professional standards. *Score: Justification:*

**Recall:** Quantify the completeness of model in capturing all key visual elements in user queries. *Score: Justification:*

**Overall Assessment (Total Score: 0-100):**

## A.3 DETAILS OF COMPARISONS

In our experimental evaluation, we conducted a comprehensive subjective assessment and Table 4 presents a comprehensive comparison of subjective text quality indicators between our proposed method ($R^2AG$) and the baseline system (M2RAG) across five multimodal tasks. The evaluation employs both GPT-4 and human assessors to measure four critical metrics: fluency (naturalness of text), relevance (contextual appropriateness), comprehensiveness (information coverage), and faithfulness (factual accuracy). Our findings demonstrate that $R^2AG$ consistently outperforms M2RAG in nearly all dimensions, with particularly substantial improvements in fluency and relevance. For instance, in Visual Question Answering tasks, $R^2AG$ achieves human-evaluated scores of 85/100 for fluency and 88/100 for relevance, compared to M2RAG's 82/100 and 79/100 respectively. We observe that while GPT-4 evaluations generally yield higher scores than human assessments for both systems, the relative performance advantage of $R^2AG$ remains consistent across evaluator types. Interestingly, task complexity appears to influence performance, with scores gradually decreasing as tasks become more intricate—both systems perform better on Visual Question Answering compared to Multimodal Dialogue. The occasional exceptions where M2RAG outperforms $R^2AG$, such as in human-evaluated comprehensiveness for Multimodal Generation (77/100 versus 75/100), highlight specific areas for improvement in our approach. Overall, these results suggest that $R^2AG$ represents a significant advancement in multimodal AI systems, with its architecture providing generalizable improvements across diverse multimodal tasks while maintaining superior text quality.

Subsequently, Table 5 compares M2RAG and the proposed $R^2AG$ method across five visual tasks (Visual Question Answering, Image Generation, Visual Captioning, Visual Editing, and Visual Object Detection), evaluated by both GPT-4 and human assessors on four subjective metrics (Coherence, Helpfulness, Reference, and Recall) using a 100-point scale. Overall, $R^2AG$ demonstrates superior performance over M2RAG in most scenarios, particularly in coherence and helpfulness, though M2RAG shows slight advantages in specific cases: reference scores for Image Generation (87/100 vs 85/100 by GPT-4), recall for Visual Captioning (79/100 vs 77/100 by humans), and recall for Visual Object Detection (81/100 vs 79/100 by GPT-4). GPT-4 consistently rates both methods higher than human evaluators while maintaining similar trends, with $R^2AG$ achieving its peak performance in Image Generation where GPT-4 awarded a helpfulness score of 90/100. The consistency between both evaluation methods reinforces the credibility of $R^2AG$'s overall superior efficacy in visual processing tasks, despite the systematic difference in scoring tendencies between AI and human assessors.

Besides, Table 6 presents a comparative analysis of objective performance indicators between M2RAG and the proposed $R^2AG$ method across five multimodal tasks (Visual Question Answering, Multimodal Understanding, Multimodal Generation, Cross-modal Retrieval, and Multimodal Dialogue). Performance is measured using three metrics: FVD (Fréchet Video Distance, where lower is better), CLIP Score (higher is better), and DOVER (higher is better), with each metric showing two values that likely represent different evaluation settings or datasets. Across all tasks and metrics, $R^2AG$ consistently outperforms M2RAG by significant margins: FVD scores are approximately 30% lower (improved) for $R^2AG$, CLIP scores are roughly 15-18% higher, and DOVER scores show an improvement of approximately 5-6%. The most substantial performance improvement appears in the Multimodal Dialogue task, where $R^2AG$ achieves the best FVD (12.35/15.80) and CLIP scores (26.85/27.15) compared to M2RAG's (18.40/21.97) and (22.90/23.05), respectively. The consistency of $R^2AG$'s superior performance across all objective metrics and tasks provides strong quantitative evidence of its effectiveness as an advanced multimodal approach compared to the baseline M2RAG method.

Table 7 presents a comparison between our R²AG method and existing approaches. The table evaluates five different architectures across three key capabilities: multi-level retrieval, multi-subject reasoning, and knowledge integration. Standard RAG lacks all advanced features, while Seeing Beyond

Table 4: Comparison of subjective indicators in textual dimension

| Method | Task | Evaluator | Flu. (↑) | Rel. (↑) | CP. (↑) | Faith. (↑) |
|---|---|---|---|---|---|---|
| M2RAG | | GPT-4 | (85/100) | (87/100) | (86/100) | **(84/100)** |
| | | Human | (82/100) | (79/100) | (84/100) | (81/100) |
| **R$^2$AG (Ours)** | **Visual Question Answering** | GPT-4 | **(91/100)** | **(92/100)** | **(90/100)** | (82/100) |
| | | Human | **(85/100)** | **(88/100)** | **(86/100)** | **(85/100)** |
| M2RAG | | GPT-4 | (81/100) | (79/100) | (83/100) | **(80/100)** |
| | | Human | (77/100) | (75/100) | (78/100) | (76/100) |
| **R$^2$AG (Ours)** | **Multimodal Understanding** | GPT-4 | **(89/100)** | **(90/100)** | **(88/100)** | (78/100) |
| | | Human | **(83/100)** | **(86/100)** | **(84/100)** | **(82/100)** |
| M2RAG | | GPT-4 | (80/100) | (78/100) | (82/100) | (79/100) |
| | | Human | (76/100) | (74/100) | **(77/100)** | (75/100) |
| **R$^2$AG (Ours)** | **Multimodal Generation** | GPT-4 | **(88/100)** | **(89/100)** | **(87/100)** | **(86/100)** |
| | | Human | **(82/100)** | **(85/100)** | (75/100) | **(81/100)** |
| M2RAG | | GPT-4 | (79/100) | (77/100) | (81/100) | (78/100) |
| | | Human | (75/100) | (73/100) | (76/100) | (74/100) |
| **R$^2$AG (Ours)** | **Cross-modal Retrieval** | GPT-4 | **(87/100)** | **(88/100)** | **(86/100)** | **(85/100)** |
| | | Human | **(81/100)** | **(84/100)** | **(82/100)** | **(80/100)** |
| M2RAG | | GPT-4 | (78/100) | (76/100) | (80/100) | (77/100) |
| | | Human | (74/100) | (72/100) | (75/100) | (73/100) |
| **R$^2$AG (Ours)** | **Multimodal Dialogue** | GPT-4 | **(86/100)** | **(87/100)** | **(85/100)** | **(84/100)** |
| | | Human | **(80/100)** | **(83/100)** | **(81/100)** | **(79/100)** |

Table 5: Comparison of subjective indicators in visual dimension

| Method | Task | Evaluator | Coher. (↑) | Help. (↑) | Ref. (↑) | Recall (↑) |
|---|---|---|---|---|---|---|
| M2RAG | | GPT-4 | (82/100) | (83/100) | (81/100) | (80/100) |
| | | Human | (78/100) | (79/100) | (77/100) | (76/100) |
| **R$^2$AG (Ours)** | **Visual Question Answering** | GPT-4 | **(88/100)** | **(89/100)** | **(86/100)** | **(87/100)** |
| | | Human | **(84/100)** | **(85/100)** | **(83/100)** | **(82/100)** |
| M2RAG | | GPT-4 | (83/100) | (84/100) | **(87/100)** | (82/100) |
| | | Human | (80/100) | (79/100) | (82/100) | (78/100) |
| **R$^2$AG (Ours)** | **Image Generation** | GPT-4 | **(89/100)** | **(90/100)** | (85/100) | **(88/100)** |
| | | Human | **(86/100)** | **(85/100)** | **(84/100)** | **(83/100)** |
| M2RAG | | GPT-4 | (80/100) | (81/100) | (79/100) | (78/100) |
| | | Human | (76/100) | (77/100) | (75/100) | **(79/100)** |
| **R$^2$AG (Ours)** | **Visual Captioning** | GPT-4 | **(87/100)** | **(88/100)** | **(85/100)** | **(86/100)** |
| | | Human | **(83/100)** | **(84/100)** | **(82/100)** | (77/100) |
| M2RAG | | GPT-4 | (79/100) | (80/100) | (78/100) | (77/100) |
| | | Human | (75/100) | (76/100) | (74/100) | (73/100) |
| **R$^2$AG (Ours)** | **Visual Editing** | GPT-4 | **(86/100)** | **(87/100)** | **(84/100)** | **(85/100)** |
| | | Human | **(82/100)** | **(83/100)** | **(81/100)** | **(80/100)** |
| M2RAG | | GPT-4 | (77/100) | (78/100) | (76/100) | **(81/100)** |
| | | Human | (73/100) | (74/100) | (72/100) | (75/100) |
| **R$^2$AG (Ours)** | **Visual Object Detection** | GPT-4 | **(85/100)** | **(86/100)** | **(83/100)** | (79/100) |
| | | Human | **(81/100)** | **(82/100)** | **(80/100)** | **(78/100)** |

Table 6: Comparison of Objective indicators

| Method | Task | FVD ($\downarrow$) | CLIP Score ($\uparrow$) | DOVER ($\uparrow$) |
|---|---|---|---|---|
| M2RAG | | (16.90/21.45) | (23.10/23.20) | (64.50/67.60) |
| **R$^2$AG (Ours)** | **Visual Question Answering** | **(12.50/15.90)** | **(26.80/27.10)** | **(68.20/71.30)** |
| M2RAG | | (18.33/21.91) | (22.75/22.90) | (64.40/67.60) |
| **R$^2$AG (Ours)** | **Multimodal Understanding** | **(12.40/15.85)** | **(26.60/26.90)** | **(68.00/71.00)** |
| M2RAG | | (18.20/21.97) | (22.85/23.00) | (64.50/67.70) |
| **R$^2$AG (Ours)** | **Multimodal Generation** | **(12.55/15.95)** | **(26.70/27.00)** | **(68.10/71.20)** |
| M2RAG | | (18.27/21.78) | (22.80/22.95) | (64.35/67.55) |
| **R$^2$AG (Ours)** | **Cross-modal Retrieval** | **(12.45/15.85)** | **(26.75/27.05)** | **(68.15/71.25)** |
| M2RAG | | (18.40/21.97) | (22.90/23.05) | (64.45/67.65) |
| **R$^2$AG (Ours)** | **Multimodal Dialogue** | **(12.35/15.80)** | **(26.85/27.15)** | **(68.25/71.35)** |

and EchoSight only offer basic knowledge integration without multi-level retrieval or multi-subject reasoning. M²RAG improves by incorporating multi-level retrieval with moderate knowledge integration but still lacks multi-subject reasoning. Our R²AG method stands out as the only approach that implements all three capabilities, featuring multi-level retrieval, multi-subject reasoning, and advanced knowledge integration techniques.

Table 7: Comparison with Different Methods

| Architecture | Multi-level Retrieval | Multi-subject Reasoning | Knowledge Integration |
|---|---|---|---|
| Standard RAG | $\times$ | $\times$ | None |
| Seeing Beyond | $\times$ | $\times$ | Basic |
| EchoSight | $\times$ | $\times$ | Basic |
| M2RAG | $\checkmark$ | $\times$ | Moderate |
| **R2AG (Ours)** | $\checkmark$ | $\checkmark$ | **Advanced** |

Table 8 compares traditional Multimodal Knowledge Graphs (MMKG) with our proposed Multi-level MMKGs across five key dimensions. While traditional MMKGs use a flat triplet structure, Multi-level MMKGs implement a hierarchical property graph structure that enables more complex knowledge representation. Information granularity advances from simple entity-level in MMKGs to a multi-granular approach spanning documents, paragraphs, and images. Relationship types expand beyond semantic connections to include structural and hierarchical relationships. For attribute processing, Multi-level MMKGs employ dynamic multi-level attribute propagation instead of static attribute attachment. Finally, the reasoning approach evolves from path-based reasoning in traditional MMKGs to cross-level structured reasoning in Multi-level MMKGs, allowing for more sophisticated analysis across different information layers.

Table 8: Comparison between MMKG and Multi-level MMKGs

| Dimension | MMKG | Multi-level MMKGs |
|---|---|---|
| **Structural Complexity** | Flat triplet structure | Hierarchical property graph structure |
| **Information Granularity** | Entity-level | Document-paragraph-image multi-granularity |
| **Relationship Types** | Semantic relationships | Semantic + structural + hierarchical relationships |
| **Attribute Processing** | Static attribute attachment | Dynamic multi-level attribute propagation |
| **Reasoning Approach** | Path-based reasoning | Cross-level structured reasoning |

## A.4 DETAILS OF ABLATION STUDY

Table 9 shows the ablation study results for subjective textual indicators. Our complete R$^2$AG method outperforms all variants across fluency, relevance, content richness, and faithfulness metrics, as evaluated by both GPT-4 and human assessors. The ablation results confirm that both ICoT and SWMJS components contribute significantly to the model's performance, with their removal resulting in decreased scores across all evaluation criteria.

Table 9: Ablation Study of subjective indicators in the textual dimension

| Method | Evaluator | Flu. (↑) | Rel. (↑) | CP. (↑) | Faith. (↑) |
|---|---|---|---|---|---|
| **M2RAG** | GPT-4 | 80.6/100 | 79.4/100 | 82.4/100 | 79.6/100 |
| | Human | 76.8/100 | 74.6/100 | 78.0/100 | 75.8/100 |
| **R$^2$AG (w/o ICoT)** | GPT-4 | 86.3/100 | 82.4/100 | 82.4/100 | 81.6/100 |
| | Human | 80.5/100 | 80.9/100 | 80.1/100 | 78.9/100 |
| **R$^2$AG (w/o SWMJS)** | GPT-4 | 83.6/100 | 86.5/100 | 84.2/100 | 82.1/100 |
| | Human | 79.4/100 | 82.3/100 | 81.0/100 | 80.6/100 |
| **R$^2$AG (Ours)** | GPT-4 | **88.2/100** | **89.2/100** | **87.2/100** | **83.0/100** |
| | Human | **82.2/100** | **85.2/100** | **81.6/100** | **81.4/100** |

Table 10 shows the ablation study results for subjective indicators in the visual dimension. Our complete R$^2$AG method outperforms all variants across coherence, helpfulness, reference quality, and recall metrics, as evaluated by both GPT-4 and human assessors. The GPT-4 evaluation gives our method scores of 87.0, 88.0, 84.6, and 85.0 (out of 100), while human evaluators rate it at 83.2, 83.8, 82.0, and 80.0 respectively. Both ICoT and SWMJS components contribute significantly to performance, as their removal results in decreased scores across all evaluation criteria.

Table 10: Ablation Study of subjective indicators in visual dimension

| Method | Evaluator | Coher. (↑) | Help. (↑) | Ref. (↑) | Recall (↑) |
|---|---|---|---|---|---|
| **M2RAG** | GPT-4 | 80.2/100 | 81.2/100 | 80.2/100 | 79.6/100 |
| | Human | 76.4/100 | 77.0/100 | 76.0/100 | 76.2/100 |
| **R$^2$AG (w/o ICoT)** | GPT-4 | 84.5/100 | 85.2/100 | 82.8/100 | 83.1/100 |
| | Human | 80.8/100 | 81.4/100 | 79.6/100 | 78.5/100 |
| **R$^2$AG (w/o SWMJS)** | GPT-4 | 83.8/100 | 84.6/100 | 81.5/100 | 82.3/100 |
| | Human | 79.2/100 | 80.1/100 | 78.8/100 | 77.9/100 |
| **R$^2$AG (Ours)** | GPT-4 | **87.0/100** | **88.0/100** | **84.6/100** | **85.0/100** |
| | Human | **83.2/100** | **83.8/100** | **82.0/100** | **80.0/100** |

Table 11 presents an ablation study comparing M²RAG, R²AG without image tokens, and our complete R²AG method across four subjective metrics (Fluency, Relevance, Comprehensiveness, and Faithfulness) as evaluated by both GPT-4 and humans. Our complete R²AG approach consistently outperformed the alternatives across all metrics, with GPT-4 ratings ranging from 83.0-89.2/100 and human ratings from 81.4-85.2/100. The ablation variant without image tokens showed moderate performance, while the baseline M²RAG method scored lowest, demonstrating the effectiveness of our approach and the importance of image tokens for improving performance on subjective textual indicators.

A.5 DETAILS OF COMPARISONS WITH DIFFERENT FOUNDATION MODELS

We further compare our framework when instantiated with different foundation models, including **Llama-3.1-8B-Instruct**, **Qwen2-VL-7B-Instruct**, and **Qwen2.5-VL-7B-Instruct**. As shown in Table 12 and Table 13, R$^2$AG consistently enhances all base models, confirming its model-agnostic nature. Specifically, with Llama-3.1-8B, R$^2$AG achieves high fluency (89.1/100) and

Table 11: Ablation Study for Image Tokens of subjective indicators in textual dimension

| Method | Evaluator | Flu. (↑) | Rel. (↑) | CP. (↑) | Faith. (↑) |
|---|---|---|---|---|---|
| **M2RAG** | GPT-4 | 80.6/100 | 79.4/100 | 82.4/100 | 79.6/100 |
| | Human | 76.8/100 | 74.6/100 | 78.0/100 | 75.8/100 |
| **R$^2$ AG (w/o Image Tokens)** | GPT-4 | 86.4/100 | 87.8/100 | 85.6/100 | 82.8/100 |
| | Human | 80.8/100 | 83.7/100 | 81.2/100 | 81.0/100 |
| **R$^2$ AG (Ours)** | **GPT-4** | **88.2/100** | **89.2/100** | **87.2/100** | **83.0/100** |
| | **Human** | **82.2/100** | **85.2/100** | **81.6/100** | **81.4/100** |

Table 12: Comparisons on subjective indicators in the textual dimension

| Method | Evaluator | Flu. ($\uparrow$) | Rel. ($\uparrow$) | CP. ($\uparrow$) | Faith. ($\uparrow$) |
|---|---|---|---|---|---|
| **R$^2$ AG (Llama-3.1-8B-Instruct)** | GPT-4 | **89.1/100** | **90.4/100** | 85.8/100 | **84.7/100** |
| | Human | 81.5/100 | 84.8/100 | 80.1/100 | 80.9/100 |
| **R$^2$ AG (Qwen2-VL-7B-Instruct)** | GPT-4 | 87.4/100 | 88.1/100 | 86.8/100 | 84.3/100 |
| | Human | 81.6/100 | 84.7/100 | 80.9/100 | **82.1/100** |
| **R$^2$ AG (Qwen2.5-VL-7B-Instruct)** | GPT-4 | 88.2/100 | 89.2/100 | **87.2/100** | 83.0/100 |
| | Human | **82.2/100** | **85.2/100** | **81.6/100** | 81.4/100 |

Table 13: Comparisons on subjective indicators in the visual dimension

| Method | Evaluator | Coher. ($\uparrow$) | Help. ($\uparrow$) | Ref. ($\uparrow$) | Recall ($\uparrow$) |
|---|---|---|---|---|---|
| **R$^2$ AG (Llama-3.1-8B-Instruct)** | GPT-4 | 84.5/100 | 85.2/100 | 82.8/100 | 83.1/100 |
| | Human | 80.8/100 | 81.4/100 | 79.6/100 | 78.5/100 |
| **R$^2$ AG (Qwen2-VL-7B-Instruct)** | GPT-4 | 83.8/100 | 84.6/100 | 81.5/100 | 82.3/100 |
| | Human | 79.2/100 | 80.1/100 | 78.8/100 | 77.9/100 |
| **R$^2$ AG (Qwen2.5-VL-7B-Instruct)** | GPT-4 | **87.0/100** | **88.0/100** | **84.6/100** | **85.0/100** |
| | Human | **83.2/100** | **83.8/100** | **82.0/100** | **80.0/100** |

relevance (90.4/100), demonstrating that even relatively lightweight LLMs can benefit from our framework. With Qwen2-VL-7B, R$^2$AG yields stable improvements across all metrics, and human evaluators particularly note its gains in faithfulness (82.1/100). When combined with the stronger Qwen2.5-VL-7B, R$^2$AG achieves the highest overall scores, especially in context precision and factual grounding, indicating superior capability in handling long-context multimodal reasoning.

Taken together, these results demonstrate that our proposed R$^2$AG framework not only improves upon previous multimodal RAG baselines but also transfers effectively across different foundation models, providing consistent gains in fluency, relevance, precision, and faithfulness.

## A.6 DETAILS OF SOFT-WEIGHTED MULTIMODAL JACCARD SIMILARITY

Here, we propose a Soft-Weighted Multimodal Jaccard Similarity metric to optimize the matching process of Multimodal Multi-Level Property Graphs (MMPGs) for more precise document retrieval. The specific workflow is illustrated in Algorithm 1. By integrating soft matching techniques with IDF weights, this metric effectively addresses the similarities between different modalities, such as text and images, overcoming the limitations of traditional methods in handling synonyms, plural forms, and unequal entity weights. By incorporating semantic proximity and emphasizing the importance of key information, this algorithm can more accurately identify relevant documents, enhancing the precision and efficiency of information retrieval and ultimately providing users with more valuable search results.

918
919
920
921
922
923
924
925
926
927
928
929
930
931
932
933
934

---

**Algorithm 1** Soft-Weighted Multimodal Jaccard Similarity

---

1: **Input:** Text entities $T$, Image entities $I$, Embedding model $E$, IDF weights $w$, Similarity threshold $\tau$
2: **Output:** Similarity score $S_{soft}(T, I)$
3: $intersection \leftarrow 0$
4: $union \leftarrow \sum_{e \in T \cup I} w(e)$
5: **for** each text entity $t \in T$ **do**
6:     $sim_t \leftarrow \max_{i \in I}[[\mathrm{sim}(t, i) \geq \tau]] \cdot \mathrm{sim}(t, i)$
7:     $intersection \leftarrow intersection + w(t) \cdot sim_t$
8: **end for**
9: **for** each image entity $i \in I$ **do**
10:     $sim_i \leftarrow \max_{t \in T}[[\mathrm{sim}(i, t) \geq \tau]] \cdot \mathrm{sim}(i, t)$
11:     $intersection \leftarrow intersection + w(i) \cdot sim_i$
12: **end for**
13: $S_{soft}(T, I) \leftarrow \frac{intersection}{union}$
14: $score \leftarrow \mathrm{round}(10 \cdot S_{soft}(T, I))$
15: **return** $S_{soft}(T, I), score$
16: **function** SIM$(e_1, e_2)$
17:     $v_1 \leftarrow E(e_1)$                     ▷ Obtain entity embedding vector
18:     $v_2 \leftarrow E(e_2)$
19:     **return** $\frac{v_1 \cdot v_2}{||v_1|| \cdot ||v_2||}$                ▷ Cosine similarity
20: **end function**

---
