# OpenReview forum: "R$^2$AG: Rethinking Retrieval-Augmented Generation via Multimodal Coherence Understanding"
_ICLR.cc/2026/Conference — ICLR 2026 Conference Withdrawn Submission_

### Official Review · Reviewer_oEr1 · 2025-10-26

**Soundness:** 2
**Presentation:** 1
**Contribution:** 2
**Rating:** 2
**Confidence:** 4

**Summary:**

This paper introduces $R^{2}AG$, a new framework for Multimodal Retrieval-Augmented Generation (MRAG) designed to improve multimodal coherence understanding. The authors argue that existing MRAG methods struggle to capture the complex structural relationships between text and visual elements. The proposed $R^{2}AG$ framework addresses this limitation through three main contributions: It represents multimodal content (like text and images) as nodes and edges in a property graph, which captures richer semantic and structural relationships than conventional embeddings. To manage the exponential complexity that arises from graph-based retrieval, the paper proposes an Implicit-CoT technique. This method partitions and analyzes local subgraphs to derive features from both node semantics and graph structure. The paper develops an enhanced graph matching algorithm that goes beyond simple feature consistency. It considers semantic approximations (like synonyms) and assigns higher priority to rare entities to improve matching robustness.

**Strengths:**

1. The authors correctly identify a standard, well-known challenge in graph-based methods: the exponential growth of complexity with additional hops. They propose the "Implicit-CoT" as a purported solution to this problem.
2. The paper includes a custom matching algorithm (Soft-Weighted Multimodal Jaccard Similarity) that attempts to account for semantic approximations, rather than relying on a standard, established graph matching technique.
3. The paper presents an extensive experimental setup, testing across five tasks with multiple metrics. While the volume of evaluation is large and includes ablation studies, the results are difficult to trust given the severe flaws in the paper's writing, formatting, and methodology.

**Weaknesses:**

1. The writing is very poor and difficult to follow. The introduction is redundant and struggles to clearly articulate the specific, high-level challenges the paper is solving.
2. The paper introduces key concepts without sufficient background. For example, it mentions a novel "graph matching algorithm", but the "Related Work" section (Section 2) provides no review of existing graph matching algorithms. This makes it impossible to judge the novelty of the proposed algorithm.
3. The "Related Work" section is largely descriptive. It lists other methods (like GraphRAG, M2RAG) but fails to provide a critical analysis of their specific "cons" or limitations. This weakens the motivation for the new framework.
4. The paper has significant formatting issues. The citation format is incorrect. Furthermore, the placement of Tables 1, 2, and 3 on pages 7 and 8 is bizarre, as they are disconnected from the "Experiment" section (Section 5) on page 8 where they are discussed. This disrupts the flow of the paper.

**Questions:**

1. Equation Clarification (Line 198): There appears to be a typo in the graph matching equation in Section 3.2. The paper states: "sim(G_Q, G_t) = alpha". This is immediately followed by "ContentSim(f_Q, f_t) + (1-alpha)StructSim(A_Q, A_t)". Was the intention that sim(G_Q, G_t) equals this weighted sum? Please provide the full, correct equation.

2. Implicit-CoT (Section 4.2): The description of the "Implicit Chain-of-Thought" (Implicit-CoT) technique is vague.
   - Step 1 (Extraction): How are the "intermediate relation paths" actually generated? Is this a separate LLM call, and if so, what specific prompts are used to guide this multi-step reasoning?
   - Step 2 (Verification): How is the confidence value V(i,j,k) generated? Is this a logit probability from the LLM, or is a separate verification model trained?
   - Step 2 (Threshold): How is the "adaptive threshold v_th" determined? Is it a fixed hyperparameter, or is it dynamically set based on the query?

---

### Official Review · Reviewer_Ceev · 2025-10-30

**Soundness:** 2
**Presentation:** 2
**Contribution:** 2
**Rating:** 4
**Confidence:** 4

**Summary:**

The paper proposes R$^2$AG, which extends multimodal RAG into a multimodal multi-layer property graph framework. It introduces an Implicit Chain-of-Thought for implicit reasoning and relation verification, combined with a Soft-Weighted Multimodal Jaccard Similarity for graph-based retrieval. Experiments demonstrate that R$^2$AG surpasses previous baselines on multiple subjective and objective metrics.

**Strengths:**

1. The paper introduces a new setting for multimodal RAG by incorporating structural information through a multimodal multi-layer property graph.

2. The evaluation contains multiple aspects, which cover both subjective dimensions (GPT-4 and human ratings on Fluency, Relevance, Coherence, Faithfulness, and visual metrics such as Coher, Help, Ref, Recall) and several objective metrics (FVD, CLIP, DOVER).

3. The proposed approach achieves SOTA performance, consistently outperforming previous baselines.

**Weaknesses:**

1. Limited novelty in introducing structural information: Incorporating structural signals into multimodal LLMs is not a new setting. Recent works such as MMAG [1], MM-GRAPH [2], and Graph4MM [3] have already explored similar ideas. The paper should include a dedicated discussion in Related Work on structural-enhanced MLLMs, directly comparing how these prior methods integrate structural information and how the proposed method differs.

[1] Multimodal Graph Learning for Generative Tasks

[2] Mosaic of Modalities: A Comprehensive Benchmark for Multimodal Graph Learning

[3] Graph4MM: Weaving Multimodal Learning with Structural Information

2. Incomplete baseline coverage: The paper overlooks several recent multimodal RAG baselines (e.g., OmniSearch [1]), which also focus on multi-step multimodal retrieval. Without comparisons to these state-of-the-art methods, it is difficult to evaluate the claimed advantages of R$^2$AG.

[1] Benchmarking Multimodal Retrieval Augmented Generation with Dynamic VQA Dataset and Self-adaptive Planning Agent

3. Scalability concerns: The proposed graph construction and verification process heavily depends on repeated calls to closed-source LLM APIs for both relation extraction and verification. As the graph size grows exponentially, the approach becomes computationally expensive and not scalable for large-scale multimodal corpora.

4. Limited dataset diversity: Experiments are conducted on only one dataset (following M2RAG). However, there exist many widely adopted multimodal RAG benchmarks such as InfoSeek [1], MRAG-Bench [2], and M^2RAG [3]. Evaluating on a single dataset fails to demonstrate the generalizability of the proposed approach.

[1] Can Pre-trained Vision and Language Models Answer Visual Information-Seeking Questions?

[2] MRAG-Bench: Vision-Centric Evaluation for Retrieval-Augmented Multimodal Models

[3] Benchmarking Retrieval-Augmented Generation in Multi-Modal Contexts

**Questions:**

Please refer to the weakness section.

There are several additional questions:

1. In line 279, does the cross-attention and feature alignment process rely on an existing vision-language model with pretrained cross-attention layers, or is it trained from scratch? If it is a new implementation, why is there no detailed formulation or architectural explanation of how this module works?

2. In the soft-weighted Jaccard similarity, how are the image entities defined? Are they extracted through object detection, segmentation, caption tags, or vision-language alignment tokens?

3. How is the IDF weighting unified across modalities so that textual and visual entities share a consistent token or feature space?

4. Why are there no reported statistics or analyses on the token cost and computational overhead during graph construction and verification, given that the method requires multiple LLM calls?

5. Has the proposed approach been tested on other closed-source or open-source LLM or VLM backbones to validate its generalizability beyond GPT4?

---

### Official Review · Reviewer_dTTR · 2025-10-31

**Soundness:** 3
**Presentation:** 2
**Contribution:** 3
**Rating:** 4
**Confidence:** 4

**Summary:**

This paper introduces R²AG, a framework that enhances multimodal Retrieval-Augmented Generation by modeling multimodal content as multi-level property graphs. The method integrates an Implicit Chain-of-Thought mechanism that leverages LLMs to infer, validate, and refine hidden semantic and cross-modal relationships within these graphs, improving structural completeness and reducing hallucination. It also proposes an enhanced graph matching algorithm that incorporates soft-weighted multimodal Jaccard similarity, accounting for semantic approximations, synonymy, and rare-entity weighting to achieve more reliable retrieval alignment. Experiments on five multimodal tasks demonstrate consistent and significant improvements over state-of-the-art baselines across both subjective evaluations (LLM- and human-based) and objective metrics for multimodal understanding and generation.

**Strengths:**

+ The design moves beyond embedding-based MRAG frameworks (e.g., M2RAG, EchoSight) by explicitly modeling multimodal documents as structured, multi-level property graphs and introducing an Implicit-CoT mechanism for implicit relation extraction, which enables fine-grained reasoning over textual and visual modalities, improving coherence and interpretability.

+ The paper presents a well-structured end-to-end framework covering multimodal preprocessing, graph construction, matching, and generation, with each component mathematically grounded and empirically validated.

+ The evaluation is across diverse multimodal tasks using both human and LLM-based assessments, providing a balanced combination of subjective and objective analyses.

**Weaknesses:**

- The method relies on several predefined hyperparameters, yet no discussion or sensitivity analysis is provided. Although Section 4.2 and Algorithm 1 describe the Soft-Weighted Multimodal Jaccard Similarity, key implementation choices (e.g, the similarity thresholding and the balancing parameter) lack theoretical or empirical justification. The paper also does not explain how the adaptive threshold $v_{th}$ handles semantic drift or ambiguity in multimodal corpora.

- The implementation and evaluation details are insufficiently described. The paper lacks clarification on model size, training cost, and experimental setup, which would help strengthen its credibility. Moreover, Table 3 only reports aggregated objective scores across the five multimodal tasks without providing per-task metric breakdowns, limiting transparency and interpretability. Also, the number of human evaluators involved in the subjective assessment is not specified.

- There is limited discussion about scalability. Given the additional property-graph construction and reasoning steps, the paper does not provide sufficient insight into whether the approach is practical for large-scale or real-time applications. Including analyses of runtime, scalability, and memory consumption would help assess its feasibility and real-world applicability. Also, it would also be helpful to report the scale of the datasets used, such as their size and composition.

**Questions:**

- As mentioned in weakness1, can the authors clarify the procedure for threshold selection ($\tau$, $v_{th}$, $\alpha$) within Algorithm 1 and the graph similarity computations? How sensitive is model performance to these parameters, and are there principled ways to select them?

- Several notations appear inconsistent or ambiguous. For instance, in Equation (2), $V$ denotes a value function, whereas in Section 3.2 it represents a node set. In addition, $f_Q$ is not formally defined, and the mapping between text/image tokens and graph nodes is vaguely specified or reused across components without clear distinction. Could the authors clarify these notational choices for better readability?

- In Sec. 4.2 Step 2, does “multimodal consistency” refer to alignment within each node’s modalities or across the connected nodes?

- The main paper presents limited evaluation results, while many detailed experiments appear only in the appendix. It would strengthen the presentation if key analyses—such as Table 9 (Ablation Study of subjective indicators in the textual dimension)—were included or summarized in the main paper for clarity and completeness.

- Regarding implementation details, what encoders and LLMs are used for graph construction? For evaluation, could the authors provide dataset statistics (e.g., dataset size, modality composition), describe the properties of the generated graphs (e.g., average number of nodes, edges, or relation types), and specify the number of people involved in the human evaluation?

---

### Official Review · Reviewer_8icw · 2025-11-01

**Soundness:** 2
**Presentation:** 2
**Contribution:** 2
**Rating:** 2
**Confidence:** 3

**Summary:**

The paper proposes R$^2$AG, a retrieval-augmented generation framework designed for multimodal and multilevel property graphs. R$^2$AG aims to enhance multimodal coherence understanding by explicitly representing relationships across text and images as attribute graphs, and by introducing Implicit Chain-of-Thought (ICoT) reasoning to manage multi-hop inference. Relevant

**Strengths:**

- The move from standard RAG to multimodal RAG has potential.
- The task is extensive, as shown in Tables 4, 5, and 6.
- The evaluation involves both human and GPT-4.

**Weaknesses:**

- The paper organization is busy, not very easy to follow, and the internal logic among the proposed contribution points is not easy to trace. For example,
  - Lines 117-130 list three major contributions, but these three do not look coherent and independent of each other.
  - The preliminary section is not easy to follow and systematically understand the whole framework, motivation, and problem setting.

- Use \citep instead of \citep
- In lines 50 to 51, the reasoning why the current RAG pipeline can not fit multimodal content is missing. The explanation in lines 54-59 is superficial.
- The RAG baseline is not adequate, and the backbone model is limited and small.

**Questions:**

Please refer to the weakness section.
- Also, why can RAGs like HippoRAG not be compared?
- How to understand the hallucination in Figure 1?

---

### Note · Authors · 2025-11-25

**Comment:**

After careful consideration of the reviewer feedback, we have decided to withdraw our submission "R²AG: Rethinking Retrieval-Augmented Generation via Multimodal Coherence Understanding" from ICLR 2026. The reviewers raised valid concerns regarding presentation clarity, incomplete baseline comparisons, and scalability issues that require substantial revisions beyond the scope of a rebuttal. We plan to address these fundamental issues and resubmit to a future venue with a significantly improved manuscript.

**Withdrawal Confirmation:**

I have read and agree with the venue's withdrawal policy on behalf of myself and my co-authors.